# Info Intervention and its Causal Calculus

**Heyang Gong**                                                                GONGHY@MAIL.USTC.EDU.CN
*University of Science and Technology of China*

**Ke Zhu**                                                                            MAZHUKE@HKU.HK
*University of Hong Kong*

**Editors:** Bernhard Schölkopf, Caroline Uhler and Kun Zhang

## Abstract

The potential outcome framework and structural causal model are two main frameworks for causal modeling, and there are efforts to combine the merits of each framework, such as the single world intervention graph (SWIG) and its potential outcome calculus. In this paper, we propose the *info* intervention inspired by understanding the causality as information transfer, and provide the corresponding causal calculus. On one hand, we explain the connection between *info* calculus and *do* calculus. On the other hand, we show that the *info* calculus is as convenient as the SWIG to check the conditional independence, and most importantly, it owns an operator $\sigma(\cdot)$ for formalizing causal queries.

**Keywords:** *info* intervention, *do* intervention, causal calculus

## 1. Introduction

Causality has been one of the basic topics of philosophy since the time of Aristotle. However during the last two decades, interest in causality has become very intense in the philosophy of science community, and a great variety of novel views on the subject have emerged and been developed. Among those novel views, it has been claimed that an informational account of causality, formally proposed and developed since Collier (1999) in philosophy, might be useful to the scientific problem of how we think about, and ultimately trace, causal linking, and so to causal inference and reasoning. An informational account of causality can be useful to help us reconstruct how science builds up understanding of the causal structure of the world. An informational account of causality may give us the prospect of saying what causality is, in a way that is not tailored to the description of reality provided by a given discipline. Moreover it carries the advantage over other causal metaphysics that it fares well with the applicability problem for other accounts of production (processes and mechanism) (Illari and Russo, 2014). But from an application in science perspective, what benefits we can gain from causal modeling? Or is it just a rhetorical flourish?

This paper explores the topic of applying information accounts of causality to the causal modeling theories, resulting in an *info* intervention framework that unifies ideas from potential outcomes and the structural causal model frameworks. In Section 2, we present some preliminaries on the major causal modeling framework. Then we propose an *info* intervention to articulate causal queries in Section 3 inspired by understanding causality as information transfer. In Section 4, we find that almost all the important results for causal DAG can be transferred to our *info* intervention framework. Specifically, we develop a causal calculus for identifying causal queries formulated by the $\sigma$ operator, proofs are offered in the Appendix A. Conclusion remarks and discussions are given in Section 5.

## 2. Preliminaries

There are many somewhat different formulations of structural causal model (SCM) in the literature, e.g., Schölkopf (2019); Pearl (2019a); Bongers et al. (2016b); Pearl et al. (2009); Forré and Mooij (2020), among which the definition in Blom et al. (2020) is used in this paper.

**Definition 1 (SCM)** *An SCM by definition consists of:*

1. *A set of nodes $V^+ = V \dot{\cup} U$, where elements of $V$ correspond to endogenous variables, elements of $U$ correspond to exogenous (or latent) variables, and $V \dot{\cup} U$ is the disjoint union of sets $V$ and $U$.*

2. *An endogenous/exogenous space $\mathcal{X}_v$ for every $v \in V^+$, and $\mathcal{X} := \prod_{v \in V^+} \mathcal{X}_v$.*

3. *A product probability measure $P := P_U = \otimes_{u \in U} P_u$ on the latent space $\prod_{u \in U} \mathcal{X}_u$.*

4. *A directed graph structure $G^+ = (V^+, E^+)$, with a set of directed edges $E^+$ and a system of structural equations $f_V = (f_v)_{v \in V}$:*

$$f_v : \prod_{s \in pa(v)} \mathcal{X}_s \to \mathcal{X}_v,$$

   *where all functions $f_V$ are measurable, and $ch(v)$ and $pa(v)$ stand for child and parent nodes of $v$ in $G^+$, respectively.*

*Conventionally, an SCM can be summarized by the tuple $M = (G^+, \mathcal{X}, P, f)$. Note that $G^+$ is referred as the augmented functional graph, while the functional graph which includes only endogenous variables, is denoted as $G$.*

According to its definition, the SCM deploys three parts, including graphical models, structural equations, and counterfactual and interventional logic. Graphical models serve as a language for representing what we know about the world, counterfactuals help us to articulate what we want to know, while structural equations serve to tie the two together in solid semantics.

Let $X_A$ be a set of variables at the nodes $A$. For any $I \subseteq V$, the key implementation of Pearl's causal diagrams is to capture interventions by using an intervention operator called $do(X_I = \tilde{x}_I)$, which simulates physical interventions by deleting certain functions from the model, replacing them with a constant vector $X_I = \tilde{x}_I$, while keeping the rest of the model unchanged. Formally, $X_I$ are called intervention variables, and this *do* intervention operator on $X_I$ is defined as follows:

**Definition 2 (*do* intervention)** *Given an SCM $M = (G^+, \mathcal{X}, P, f)$ for $X_V$ and any $I \subseteq V$, the do intervention $do(X_I = \tilde{x}_I)$ (or, in short, $do(\tilde{x}_I)$) maps $M$ to the do-intervention model $M^{do(\tilde{x}_I)} = (G^+, \mathcal{X}, P, \tilde{f})$ for $X_V$, where*

$$\tilde{f}_v(X_{pa(v) \cap V}, X_{pa(v) \cap U}) := \begin{cases} \tilde{x}_v, & v \in I, \\ f_v(X_{pa(v) \cap V}, X_{pa(v) \cap U}), & v \in V \setminus I. \end{cases}$$

There are theoretical and technical complications in dealing with the cyclic SCM (Bongers et al., 2016a), and currently most of literature concentrates on the case with the acyclic SCMs which associates with a causal directed acyclic graph (DAG).

Generally speaking, the DAG can be viewed as the non-parametric analogue of an acyclic SCM. Denote a DAG by $G = (V, E)$, with a set of nodes $V$ and a set of directed edges $E$. For ease of notation, we write $X := X_V$ as the variables at $V$. To do causal inference in $G$, we need specify a way to calculate the intervention distribution of $X$ in the *do*-intervention DAG, and this leads to the so-called causal DAG, under which the causal semantics could be well defined without any complications.

**Definition 3 (Causal DAG)** *Consider a DAG $G = (V, E)$ and a random vector $X$ with distribution $P$. Then, $G$ is called a causal DAG for $X$ if $P$ satisfies the following:*

1. *$P$ factorizes, and thus is Markov, according to $G$, and*

2. *for any $A \subseteq V$, $B = V/A$, and any $\tilde{x}_A, x_B$ in the domains of $X_A, X_B$,*

$$P(x|do(\tilde{x}_A)) = \prod_{k \in B} P(x_k|x_{pa(k)}) \prod_{j \in A} \mathbb{I}(x_j = \tilde{x}_j). \tag{1}$$

In view of (1), the difference between the observational distribution $P(x)$ and the *do*-intervention distribution $P(x|do(\tilde{x}_A))$ is that all factors $P(x_j|x_{pa(j)})$, $j \in A$, are removed and replaced by degenerate probabilities $\mathbb{I}(x_j = \tilde{x}_j)$, while all remaining factors $P(x_k|x_{pa(k)})$, $k \in B$, stay the same.

For a causal DAG $G$, its *do*-intervention DAG is defined as follows:

**Definition 4 (*do*-intervention DAG)** *Consider a causal DAG $G = (V, E)$ for a random vector $X$, and its do intervention $do(\tilde{x}_A)$. Then, the do-intervention DAG, denoted by $G^{do(\tilde{x}_A)}$, is for $X$, which has the do-intervention distribution $P(x|do(\tilde{x}_A))$ in (1).*

Comparing to the SCM framework which uses deterministic functions to denote causal mechanisms, the potential outcome framework is an intuitive experimental causality approach. The starting point is a population of units. There are then three components of the potential outcomes approach. First, there is a treatment/cause that can take on different values for each unit. Each unit in the population is characterized by a set of potential outcomes $Y(x)$, one for each level of the treatment. Second, the causal effects correspond to comparisons of the potential outcomes, of which at most one can be observed, with all the others missing. Paul Holland refers to this as the "fundamental problem of causal inference," (Holland (1986), p. 59). The third key component is the assignment mechanism that determines which units receive which treatments. Many efforts have been made to combine these two frameworks, such as the single-world intervention graph (SWIG) (Richardson and Robins, 2013) and the po-calculus (Malinsky et al., 2019). In the next section, we start to introduce our *info* intervention framework inspired by information accounts of causality, which can be considered as a connection between the two frameworks.

## 3. Info Intervention

The view of understanding as information transfer is first formally proposed by Collier (1999), and more details on information accounts of causality can be found in Illari and Russo (2014). Unlike the *do* intervention that is a modification of causal mechanisms, the *info* intervention proposed below modifies the output information of a variable.

**Definition 5 (*Info* intervention)**  *Given an SCM* $M = (G^+, \mathcal{X}, P, f)$ *for* $X_V$ *and any* $I \subseteq V$, *the info intervention* $\sigma(X_I = \tilde{x}_I)$ *(or, in short,* $\sigma(\tilde{x}_I)$*) maps* $M$ *to the info-intervention model* $M^{\sigma(\tilde{x}_I)} = (G^+, \mathcal{X}, P, f)$ *for* $X_V^{\sigma(\tilde{x}_I)}$, *where*

$$X_v^{\sigma(\tilde{x}_I)} = f_v(\widetilde{X}_{V \cap pa(v)}, X_{U \cap pa(v)})$$

*with* $\widetilde{X}_j = \tilde{x}_j$ *if* $j \in I$ *else* $X_j^{\sigma(\tilde{x}_I)}$.

Let $desc(I)$ denote the descendant nodes of every node in $I$. Based on definition 5, we can show that for any node $i \notin desc(A)$ with $A \subseteq V$, $X_i^{\sigma(\tilde{x}_A)} = X_i$. Also, for two disjoint sets $A, B \subseteq V$, $X_v^{\sigma(\tilde{x}_A, \tilde{x}_B)} := \left(X_v^{\sigma(\tilde{x}_A)}\right)^{\sigma(\tilde{x}_B)}$ has the commutative property, that is, $X_v^{\sigma(\tilde{x}_A, \tilde{x}_B)} = X_v^{\sigma(\tilde{x}_B, \tilde{x}_A)}$ for all $v \in V$. Moreover, based on Definition 5, we know that the *info*-intervention SCM $M^{\sigma(\tilde{x}_I)}$ does not delete any structural equations $f_V$ from the model, but just sends out the information $X_I = \tilde{x}_I$ to $desc(I)$. Since the information $X_I = \tilde{x}_I$ has been received by $desc(I)$, the edges from $I$ to $ch(I)$ (i.e., the child nodes of $I$) are removed in $M^{\sigma(\tilde{x}_I)}$.

To further illustrate how the *info* intervention works and what the differences are between *info* and *do* interventions , we consider the following example:

**Example 1**  *Consider an SCM* $M$ *with a treatment* $T$, *an outcome* $Y$, *a confounder* $Z$, *and two latent variables* $\epsilon_T, \epsilon_Z$, *where its structural equations are specified as follows:*

$$\begin{cases} Z = f_Z(\epsilon_Z), \\ T = f_T(Z, \epsilon_T), \\ Y = f_Y(T, Z). \end{cases}$$

*Based on Definition 2, its do-intervention SCM* $M^{do(\tilde{t})}$ *has the following structural equations:*

$$\begin{cases} Z = f_Z(\epsilon_Z), \\ T = \tilde{t}, \\ Y = f_Y(T, Z). \end{cases}$$

*Based on Definition 5, its info-intervention SCM* $M^{\sigma(\tilde{t})}$ *has the following structural equations:*

$$\begin{cases} Z = f_Z(\epsilon_Z), \\ T = f_T(Z, \epsilon_T), \\ Y^{\sigma(\tilde{t})} = f_Y(\tilde{t}, Z), \end{cases}$$

*where we have used the fact that* $Z^{\sigma(\tilde{t})} = Z$ *and* $T^{\sigma(\tilde{t})} = T$. *Note that the form of causal mechanisms (i.e.,* $f_Z$, $f_T$ *and* $f_Y$*) are unchanged only in* $M^{\sigma(\tilde{t})}$.

There is only partial knowledge of the underlying SCM is available in most practical settings, and then graphical causal models such as causal DAG are widely used for causal modeling. Analogous to the causal DAG, it is natural to study the causal relationship in $X$ by introducing our *info*-causal DAG below, under which the causal semantics could be well defined without any complications in the framework of *info* intervention.

**Definition 6 (*Info*-causal DAG)** *Consider a DAG $G = (V, E)$ and a random vector $X$ with distribution $P$. Then, $G$ is called an info-causal DAG for $X$ if $P$ satisfies the following:*

*1. $P$ factorizes, and thus is Markov, according to $G$,*

*2. for any $A \subseteq V$ and any $\tilde{x}_A$ in the domains of $X_A$,*

$$P(x|\sigma(\tilde{x}_A)) = \prod_{k \in V} P(x_k|x^*_{pa(k)}), \qquad (2)$$

*where $x^*_k = x_k$ if $k \notin A$ else $\tilde{x}_k$.*

In view of (2), the difference between the observational distribution $P(x)$ and the *info*-intervention distribution $P(x|\sigma(\tilde{x}_A))$ is that the factors $P(x_k|x_{pa(k)})$ satisfying $pa(k) \cap A \neq \emptyset$ in $P(x)$, are replaced by $P(x_k|x^*_{pa(k)})$ in $P(x|\sigma(\tilde{x}_A))$ with $x_j, j \in pa(k) \cap A$, replaced by $\tilde{x}_j$, while all remaining factors $P(x_k|x_{pa(k)})$ satisfying $pa(k) \cap A = \emptyset$ in $P(x)$, are unchanged after *info* intervention.

For a causal DAG, its *do*-intervention DAG is closely related to graphical rules for identification of causal queries. For our *info*-causal DAG $G$, its *info*-intervention DAG below can not only work as the *do*-intervention DAG, but also provide a graphical way to check the conditional independence between factual and counterfactual variables.

**Definition 7 (*Info*-intervention DAG)** *Consider an info-causal DAG $G = (V, E)$ for a random vector $X$, and its info intervention $\sigma(\tilde{x}_A)$. The info-intervention DAG, denoted by $G^{\sigma(\tilde{x}_A)}$, is for $X^{\sigma(\tilde{x}_A)}$, which has the info-intervention distribution $P(x|\sigma(\tilde{x}_A))$ in (2), where $X^{\sigma(\tilde{x}_A)}$ is defined in the same way as $X$, except that the variables at descendant nodes of $A$ (say, $X_{desc(A)}$) are replaced by the counterfactual variables (say, $X^{\sigma(\tilde{x}_A)}_{desc(A)}$).*

Due to the distinct forms of intervention distribution, our *info*-intervention DAG $G^{\sigma(\tilde{x}_A)}$ has some differences from Pearl's causal graphs. To further illustrate these differences graphically, we consider the following example:

**Example 2** *A DAG $G$ with four disjoint sets of variables $X_A$, $X_B$, $X_C$ and $X_D$ is given in Fig. 1(a). Take $X_A$ as the intervention variables. Then, the do-intervention DAG $G^{do(\tilde{x}_A)}$ (see Fig. 1(b)) removes the arrows from $pa(A)$ to $A$, and forces the intervention variables $X_A$ to take the hypothetical values $\tilde{x}_A$. On the contrary, the info-intervention DAG $G^{\sigma(\tilde{x}_A)}$ (see Fig. 1(c)) removes the arrows from $A$ to $ch(A)$, and forces each variable $X_i$ to be $X^{\sigma(\tilde{x}_A)}_i$, where the variable $X^{\sigma(\tilde{x}_A)}_i$ is a counterfactual variable if $i \in desc(A)$. In other words, $X^{\sigma(\tilde{x}_A)}_D = X_D$ are not counterfactual variables, $X^{\sigma(\tilde{x}_A)}_B$ and $X^{\sigma(\tilde{x}_A)}_C$ are always counterfactual variables, and the variable $X^{\sigma(\tilde{x}_A)}_i$, $i \in A$, is a counterfactual variable if $i \in desc(A)$.*

## 4. Causal Calculus for Info Intervention

The *do* calculus is one of the most important merits for the Pearl's causal diagrams, and we develop a corresponding causal calculus for info intervention in this section. Denote by $\perp\!\!\!\perp_d$ the $d$-separation,

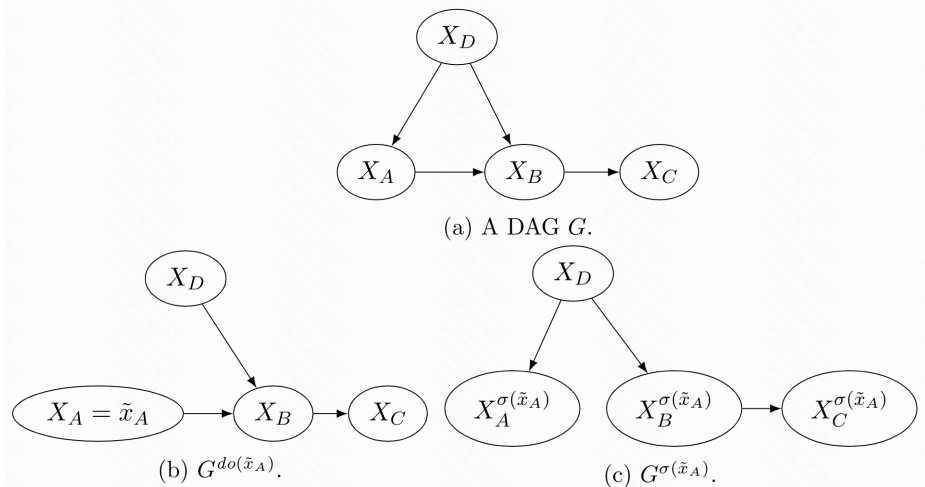

(a) A DAG $G$.

(b) $G^{do(\tilde{x}_A)}$.

(c) $G^{\sigma(\tilde{x}_A)}$.

Figure 1: A DAG and its two intervention DAGs.

$anc(I)$ the ancestor nodes of every node in $I$, and $G_{\bar{I}}$ the graph obtained by deleting from $G$ all arrows pointing to nodes in $I$. For ease of presentation, the following abbreviations are used below:

$$P(x_B|\sigma(\tilde{x}_A)) := P(X_B^{\sigma(\tilde{x}_A)} = x_B) \text{ in } G^{\sigma(\tilde{x}_A)},$$

$$P(x_B|\sigma(\tilde{x}_A), x_C) := \frac{P(x_B, x_C|\sigma(\tilde{x}_A))}{P(x_C|\sigma(\tilde{x}_A))}.$$

Similar to Causal DAG, we have "Back-door/Front-door" criteria for *info*-causal DAG to identify causal queries formulated by *info* intervention.

**Theorem 8** *For an info-causal DAG $G$ in Fig. 2,*

$$P(x_B|\sigma(\tilde{x}_A)) = \sum_{x_C} P(x_B|\tilde{x}_A, x_C)P(x_C),$$

*where all the back-door paths from $A$ to $B$ are blocked by $C$.*

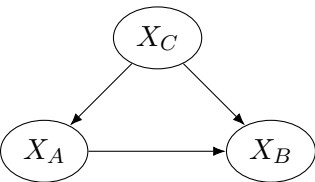

Figure 2: An *info*-causal DAG satisfying the "back-door" criterion.

**Theorem 9** *For an info-causal DAG $G$ in Fig. 3,*

$$P(x_B|\sigma(\tilde{x}_A)) = \sum_{x_C} P(x_C|\tilde{x}_A) \sum_{x_A} P(x_B|x_C, x_A)P(x_A),$$

*where 1) $C$ intercepts all paths from $A$ to $B$; 2) there is no unblocked back-door path from $A$ to $C$; and 3) all back-door paths from $C$ to $B$ are blocked by $A$.*

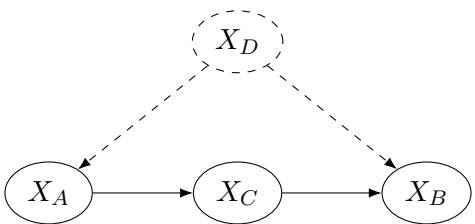

Figure 3: An *info*-causal DAG satisfying the "front-door" criterion.

[Pearl](1995) provided 3 rules for *do* intervention, which enable us to identify a causal query and turn a causal question into a statistical estimation problem. Specifically, Pearl's 3 rules describe graphical criteria for

1. insertion/deletion of observations,
2. action/observation exchange,
3. insertion/deletion of actions.

By using those three rules, the expression of *do* intervention probability may be reduced step-wisely to an equivalent expression involving only observational probabilities. Similar to Pearl's 3 rules for *do* intervention, we can present our three rules for *info* intervention.

**Theorem 10 (Three rules for *info* intervention)** *For an info-causal DAG $G$, $A, B, C$ and $D$ are its arbitrary disjoint node sets. Then,*
    **Rule 1** *(Insertion/deletion of observations)*
$$P(x_B|\sigma(\tilde{x}_A), x_C, x_D) = P(x_B|\sigma(\tilde{x}_A), x_D) \text{ if } B \perp\!\!\!\perp_d C|D \text{ in } G^{\sigma(\tilde{x}_A)};$$
    **Rule 2** *(Action/observation exchange)*
$$P(x_B|\sigma(\tilde{x}_A), \sigma(\tilde{x}_C), x_D) = P(x_B|\sigma(\tilde{x}_A), \tilde{x}_C, x_D) \text{ if } B \perp\!\!\!\perp_d C|D \text{ in } G^{\sigma(\tilde{x}_A, \tilde{x}_C)};$$
    **Rule 3** *(Insertion/deletion of actions)*
$$P(x_B|\sigma(\tilde{x}_A), \sigma(\tilde{x}_C), x_D) = P(x_B|\sigma(\tilde{x}_A), x_D) \text{ if } B \perp\!\!\!\perp_d C|D \text{ in } G^{\sigma(\tilde{x}_A)}_{\overline{C/anc(D)}},$$
*where $C/anc(D)$ is the set of $C$-nodes that are not ancestors of any $D$-node* [1].

In some applications, we may use the following simpler version of three rules in Theorem 10.

**Theorem 11** *For an info-causal DAG $G$, $A, B, C$ and $D$ are its arbitrary disjoint node sets. Then,*
    **Rule 1** *(Insertion/deletion of observations)*
$$P(x_B|\sigma(\tilde{x}_A), x_C, x_D) = P(x_B|\sigma(\tilde{x}_A), x_D) \text{ if } B \perp\!\!\!\perp_d C|D \text{ in } G^{\sigma(\tilde{x}_A)};$$
    **Rule 2** *(Action/observation exchange)*
$$P(x_B|\sigma(\tilde{x}_A), x_C) = P(x_B|\tilde{x}_A, x_C) \text{ if } B \perp\!\!\!\perp_d A|C \text{ in } G^{\sigma(\tilde{x}_A)};$$
    **Rule 3** *(Insertion/deletion of actions)*
$$P(x_B|\sigma(\tilde{x}_A)) = P(x_B) \text{ if there are no causal paths from } A \text{ to } B \text{ in } G.$$

In view of Theorems 10 and 11, we can see that our formulas are similar to those in *do* calculus. This is because the distributions of all non-intervention variables are the same in both *info*- and *do*-intervention DAGs, and the random intervention variables in *info*-intervention DAG behave similarly as the deterministic intervention variables in *do*-intervention DAG, due to the fact that the

---

1. The notation of $G_{\bar{A}}$ is used to denote when arrows directed into the set $A$ have been removed from the graph $G$.

intervention variables in *info*-intervention DAG with only possible converging arrows do not cause any other variables. Indeed, by (1) and (2), it is straightforward to see that for arbitrary disjoint node sets $A, B$ and $C$ in $V$,

$$P(x_B|do(\tilde{x}_A), x_C) = P(x_B|\sigma(\tilde{x}_A), x_C). \tag{3}$$

The result (3) implies that our formulas on three Rules in Theorem 10 are also the same as Pearl's formulas on three Rules in Pearl (1995).

The result (3) also indicates that Pearl's causal calculus and our causal calculus are exchangeable, but this does not mean the same manipulating convenience in both frameworks. Theorem 12 below shows that our conditions for checking Rules 1–3 in Theorem 10 are equivalent to those for checking Rules 1–3 in Pearl (1995), and they tend to be more convenient for use since the intervention nodes $A$ are not involved as part of conditioning set in *info*-intervention DAG.

**Theorem 12 (Equivalence of checking conditions)** *For an info-causal DAG $G$, $A, B, C$ and $D$ are its arbitrary disjoint node sets. Then,*

*(i) $B \perp\!\!\!\perp_d C|D$ in $G^{\sigma(\tilde{x}_A)} \iff B \perp\!\!\!\perp_d C|A, D$ in $G_{\overline{A}}$;*

*(ii) $B \perp\!\!\!\perp_d C|D$ in $G^{\sigma(\tilde{x}_A, \tilde{x}_C)} \iff B \perp\!\!\!\perp_d C|A, D$ in $G_{\overline{A}}^{\sigma(\tilde{x}_C)}$;*

*(iii) $B \perp\!\!\!\perp_d C|D$ in $G_{\overline{C/anc(D)}}^{\sigma(\tilde{x}_A)} \iff B \perp\!\!\!\perp_d C|A, D$ in $G_{\overline{A}, \overline{C/anc(D)}}$.*

Denote $X_A \perp\!\!\!\perp X_B|X_C$ by the conditional independence of $X_A$ and $X_B$, given $X_C$. To end this section, we re-visit an example in Richardson and Robins (2013).

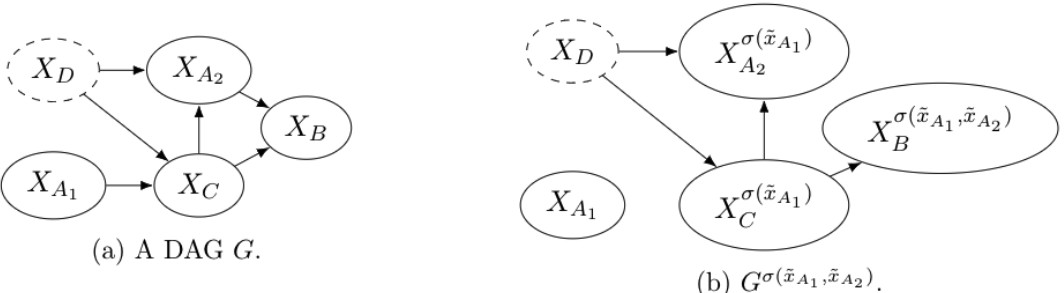

(a) A DAG $G$.

(b) $G^{\sigma(\tilde{x}_{A_1}, \tilde{x}_{A_2})}$.

Figure 4: A DAG $G$ and its *info*-intervention DAG.

**Example 3** *Consider a DAG $G$ in Fig. 4(a), where $i \notin desc(A_1)$ for any $i \in A_1$, and $i \notin desc(A_2)$ for any $i \in A_2$. Fig. 4(b) plots the info-intervention DAG $G^{\sigma(\tilde{x}_{A_1}, \tilde{x}_{A_2})}$, where we have used the fact*

$$X_{A_1}^{\sigma(\tilde{x}_{A_1}, \tilde{x}_{A_2})} = \left(X_{A_1}^{\sigma(\tilde{x}_{A_1})}\right)^{\sigma(\tilde{x}_{A_2})} = \left(X_{A_1}\right)^{\sigma(\tilde{x}_{A_2})} = X_{A_1},$$

$$X_{A_2}^{\sigma(\tilde{x}_{A_1}, \tilde{x}_{A_2})} = \left(X_{A_2}^{\sigma(\tilde{x}_{A_2})}\right)^{\sigma(\tilde{x}_{A_1})} = X_{A_2}^{\sigma(\tilde{x}_{A_1})},$$

$$X_C^{\sigma(\tilde{x}_{A_1}, \tilde{x}_{A_2})} = \left(X_C^{\sigma(\tilde{x}_{A_1})}\right)^{\sigma(\tilde{x}_{A_2})} = X_C^{\sigma(\tilde{x}_{A_1})}, \ X_D^{\sigma(\tilde{x}_{A_1}, \tilde{x}_{A_2})} = X_D.$$

*Then, $X_B^{\sigma(\tilde{x}_{A_1}, \tilde{x}_{A_2})} \perp\!\!\!\perp X_{A_2}^{\sigma(\tilde{x}_{A_1})}|X_{A_1}, X_C^{\sigma(\tilde{x}_{A_1})}$, since $B \perp\!\!\!\perp_d A_2|A_1, C$ in Fig. 4(b). Note that this conclusion was also proved in Richardson and Robins (2013) by constructing an SWIG.*

*Besides the checking of independence between counterfactual variables, we can also calculate* $P(x_B|\sigma(\tilde{x}_{A_1}, \tilde{x}_{A_2}), x_C)$ *(i.e., the conditional probability of counterfactual variables* $X_B^{\sigma(\tilde{x}_{A_1}, \tilde{x}_{A_2})}$ *given* $X_C^{\sigma(\tilde{x}_{A_1}, \tilde{x}_{A_2})}$*) by*

$$P(x_B|\sigma(\tilde{x}_{A_1}, \tilde{x}_{A_2}), x_C)$$
$$= P(x_B|\sigma(\tilde{x}_{A_1}), \tilde{x}_{A_2}, x_C) \text{ (by Rule 2 in Theorem 10)}$$
$$= P(x_B|\tilde{x}_{A_2}, x_C) \text{ (by Rule 3 in Theorem 10)}.$$

Finally, we shall mention that our *info*-causal DAG is closely related to the SWIG. The SWIG is an approach to unifying graphs and counterfactuals via splitting every intervention node into a random node and a fixed node, and its causal calculus can be implemented by using po-calculus (Malinsky et al., 2019) as follows:

$$p(Y(x) \mid Z(x), W(x)) = p(Y(x) \mid W(x)), \quad \text{if } (Y(x) \perp\!\!\!\perp Z(x) \mid W(x))_{\mathcal{G}(x)};$$

$$p(Y(x, z) \mid W(x, z)) = p(Y(x) \mid W(x), Z(x) = z), \quad \text{if } (Y(x, z) \perp\!\!\!\perp Z(x, z) \mid W(x, z))_{\mathcal{G}(x, z)};$$

$$p(Y(x, z) \mid W(x, z)) = p(Y(x) \mid W(x)), \quad \text{if } \begin{cases} (Y(x, z_1), W(x, z_1) \perp\!\!\!\perp z_1)_{\mathcal{G}(x, z_1)}, \\ (Y(x, z_1) \perp\!\!\!\perp Z_2(x, z_1) \mid W(x, z_1))_{\mathcal{G}(x, z_1)}, \end{cases}$$

where $Z_1$ is set of $Z$-nodes that are not ancestors of any $W$-node and $Z_2 = Z \setminus Z_1$, and $\mathcal{G}(x)$, $\mathcal{G}(x, z), \mathcal{G}(x, z_1)$ are corresponding SWIGs. Obviously, the graphical criteria for po-calculus is different from our causal calculus for *info* or *do* intervention, since its Rule 3 relies on two $d$-separation conditions on the SWIG $\mathcal{G}(x, z_1)$ instead of one. Moreover, the main feature of our causal calculus for *info*-causal DAG is the use of intervention idea via the novel operator $\sigma(\cdot)$, whereas the po-calculus for SWIG does not have this feature. Finally, the potential outcomes framework reflects the view of experimental causality, while *info* intervention framework intuitively describes an action that intervenes the information transferred among variables which reflects the informational accounts of causality.

## 5. Concluding Remarks and Discussions

There are many causal notations and tools across different disciplines. Though the SCM and potential outcomes are currently the most popular frameworks used by causality researchers, still we might benefit from other tools, such as various philosophical accounts of causality. Inspired by ideas from the information accounts of causality, we propose an *info* intervention framework based on structural causal model, and develop the corresponding graphical models with a causal calculus. We not only explain the connection between *info* calculus and others, but also introduce an operator $\sigma(\cdot)$ for formalizing causal queries. We feel that the informational accounts of causality could be further explored to complement the current causal modeling theories.

**Discussions of future research directions.** The algorithmic information theory has already been used in causal modeling (see e.g. Schölkopf (2019)), and there is exploration of causal emergence with information-theoretic measures (Dewhurst, 2021). It is also worth mentioning that, broadly speaking, the information accounts of causality can also facilitate interpretation of existing widely used causal propositions. For example, regarding back-door/front-door criteria, the goal of

which can be consistently considered as whether the observational information of a set of variables is enough or not to answer causal-effect estimation question, instead of conventional understanding in terms of controlling variables. Moreover, in general sense of information accounts, Pearl points out that questions in one layer of the causal hierarchy can only be answered when corresponding layer information is available (Pearl, 2019b; Bareinboim et al., 2020), and Scholköpf believes causal science will enable AI systems to act and make decisions with information from Lorenzian imagined space (Schölkopf, 2019). The recently proposed Mini-Turing test for AI — How can machines represent causal knowledge in a way that would enable them to access the necessary information swiftly, answer questions correctly, and do it with ease, as a human can? (Pearl and Mackenzie, 2018). In summary, to build true intelligent machines, climb the ladder of data, information, knowledge and wisdom, we might need to incorporate the information accounts of causality[2] into causal tasks.

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

## Appendix A. Proofs

To facilitate our proofs, we give a technical lemma:

**Lemma 13** *For an info-causal DAG $G$, $A, B$ and $C$ are its arbitrary disjoint node sets. Then,*
   *(i) $P(x|\sigma(x_A)) = P(x)$;*
   *(ii) $P(x_A, x_B|\sigma(x_A)) = P(x_A, x_B)$;*
   *(iii) $P(x_B|x_A, \sigma(x_A)) = P(x_B|x_A)$;*
   *(iv) $P(x_A, x_B|\sigma(x_A), \sigma(x_C)) = P(x_A, x_B|\sigma(x_C))$;*
   *(v) $P(x_B|x_A, \sigma(x_A), \sigma(x_C)) = P(x_B|x_A, \sigma(x_C))$.*

**Proof of Lemma 13.** The result (i) holds by taking $\tilde{x}_A = x_A$ in (2). By (i) and the marginalization over $x_A \cup x_B$ and $x_A$, it follows that

$$P(x_A, x_B|\sigma(x_A)) = P(x_A, x_B) \quad \text{and} \quad P(x_A|\sigma(x_A)) = P(x_A),$$

which entail the results (ii)–(iii). By (i), we know that $P(x|\sigma(x_A), \sigma(x_C)) = P(x|\sigma(x_C)) = P(x)$, which entails the result (iv) by the marginalization over $x_A \cup x_B$. Finally, the result (v) holds by (iv) and a similar argument as for (iii). This completes all of the proofs.

**Proof of Theorem 8.** By (2), we have

$$P(x_B, x_A, x_C|\sigma(\tilde{x}_A)) = P(x_B|\tilde{x}_A, x_C)P(x_A|x_C)P(x_C),$$

which entails

$$
\begin{aligned}
P(x_B|\sigma(\tilde{x}_A)) &= \sum_{x_C}\sum_{x_A} P(x_B, x_A, x_C|\sigma(\tilde{x}_A)) \\
&= \sum_{x_C}\sum_{x_A} P(x_B|\tilde{x}_A, x_C)P(x_A|x_C)P(x_C) \\
&= \sum_{x_C} P(x_B|\tilde{x}_A, x_C)P(x_C) \sum_{x_A} P(x_A|x_C) \\
&= \sum_{x_C} P(x_B|\tilde{x}_A, x_C)P(x_C).
\end{aligned}
$$

This completes the proof.

**Proof of Theorem 9.** By (2), we have

$$
P(x_B, x_C, x_A, x_D|\sigma(\tilde{x}_A)) = P(x_B|x_D, x_C)P(x_C|\tilde{x}_A)P(x_A|x_D)P(x_D).
$$

Moreover, it is easy to see that $C \perp\!\!\!\perp_d D|A$ and $A \perp\!\!\!\perp_d B|C, D$ in Fig. 3. In $G$, since the $d$-separation implies the conditional independence, we know that $X_C$ and $X_D$ are independent given $X_A$, and $X_A$ and $X_B$ are independent given $X_C$ and $X_D$. Hence,

$$
P(x_C|x_D, x_A) = P(x_C|x_A) \text{ and } P(x_B|x_D, x_C) = P(x_B|x_D, x_C, x_A), \tag{4}
$$

where the first equality further implies

$$
P(x_D|x_C, x_A) = P(x_D|x_A). \tag{5}
$$

Then, it follows that

$$
\begin{aligned}
P(x_B&|\sigma(\tilde{x}_A)) \\
&= \sum_{x_C}\sum_{x_A}\sum_{x_D} P(x_B, x_C, x_A, x_D|\sigma(\tilde{x}_A)) \\
&= \sum_{x_C}\sum_{x_A}\sum_{x_D} P(x_B|x_D, x_C)P(x_C|\tilde{x}_A)P(x_A|x_D)P(x_D) \\
&= \sum_{x_C} P(x_C|\tilde{x}_A) \sum_{x_A}\sum_{x_D} P(x_B|x_D, x_C)P(x_A|x_D)P(x_D) \\
&= \sum_{x_C} P(x_C|\tilde{x}_A) \sum_{x_A}\sum_{x_D} P(x_B|x_D, x_C, x_A)P(x_D|x_A)P(x_A) \text{ by (4)} \\
&= \sum_{x_C} P(x_C|\tilde{x}_A) \sum_{x_A} P(x_A) \sum_{x_D} P(x_B|x_D, x_C, x_A)P(x_D|x_A) \\
&= \sum_{x_C} P(x_C|\tilde{x}_A) \sum_{x_A} P(x_A) \sum_{x_D} P(x_B|x_D, x_C, x_A)P(x_D|x_C, x_A) \text{ by (5)} \\
&= \sum_{x_C} P(x_C|\tilde{x}_A) \sum_{x_A} P(x_A)P(x_B|x_C, x_A).
\end{aligned}
$$

This completes the proof.

To prove Theorem 10, we first prove Theorem 11.

**Proof of Theorem 11.** In $G^{\sigma(\tilde{x}_A)}$, since $P(x_A, x_B, x_C, x_D|\sigma(\tilde{x}_A))$ factorizes, the $d$-separation implies the conditional independence (Geiger, Verma and Pearl, 1990).

For Rule 1, we know that $X_B$ and $X_C$ are independent given $X_D$ in $G^{\sigma(\tilde{x}_A)}$, and hence the conclusion holds.

For Rule 2, since $X_B$ and $X_A$ are independent given $X_C$ in $G^{\sigma(\tilde{x}_A)}$, we have that $P(x_B|\sigma(\tilde{x}_A), x_C) = P(x_B|\sigma(\tilde{x}_A), \tilde{x}_A, x_C)$. Then, the conclusion holds since

$$P(x_B|\sigma(\tilde{x}_A), \tilde{x}_A, x_C) = \frac{P(x_B, \tilde{x}_A, x_C|\sigma(\tilde{x}_A))}{P(\tilde{x}_A, x_C|\sigma(\tilde{x}_A))} = \frac{P(x_B, \tilde{x}_A, x_C)}{P(\tilde{x}_A, x_C)},$$

where the second equality holds by Lemma 13(ii).

For Rule 3, let $Anc(B) = anc(B) \cup B$. Then, by (2), we have

$$\begin{aligned}
P(x|\sigma(\tilde{x}_A)) &= \prod_{k \in V} P(x_k|x^*_{pa(k)}) \\
&= \prod_{k \in Anc(B)} P(x_k|x^*_{pa(k)}) \cdot \prod_{k \notin Anc(B)} P(x_k|x^*_{pa(k)}) \\
&= \prod_{k \in Anc(B)} P(x_k|x_{pa(k)}) \cdot \prod_{k \notin Anc(B)} P(x_k|x^*_{pa(k)}),
\end{aligned}$$

where we have used the fact that $x^*_{pa(k)} = x_{pa(k)}$ for any $k \in Anc(B)$, since $A \cap Anc(B) = \emptyset$. Marginalizing over $x_{Anc(B)}$, we can obtain

$$P(x_{Anc(B)}|\sigma(\tilde{x}_A)) = \prod_{k \in Anc(B)} P(x_k|x_{pa(k)}) = P(x_{Anc(B)}).$$

Since $B \in Anc(B)$, the conclusion follows directly. This completes all of the proofs.

For Theorem 10, its Rule 1 has been proved in Theorem 11, and its Rules 2 and 3 are proved below.

**Proof of Theorem 10** (Rule 2). Since $X_B$ and $X_C$ are independent given $X_D$ in $G^{\sigma(\tilde{x}_A, \tilde{x}_C)}$, we have that $P(x_B|\sigma(\tilde{x}_A, \tilde{x}_C), x_D) = P(x_B|\sigma(\tilde{x}_A, \tilde{x}_C), \tilde{x}_C, x_D)$. Then, the conclusion holds since

$$\begin{aligned}
P(x_B|\sigma(\tilde{x}_A, \tilde{x}_C), \tilde{x}_C, x_D) &= \frac{P(x_B, x_D|\sigma(\tilde{x}_A, \tilde{x}_C), \tilde{x}_C)}{P(x_D|\sigma(\tilde{x}_A, \tilde{x}_C), \tilde{x}_C)} \\
&= \frac{P(x_B, x_D|\sigma(\tilde{x}_A), \tilde{x}_C)}{P(x_D|\sigma(\tilde{x}_A), \tilde{x}_C)},
\end{aligned}$$

where the last equality holds by Lemma 13(v).

To prove Rule 3 in Theorem 10, we need an additional lemma.

**Lemma 14** *For an info-causal DAG $G$, $B$, $C_1$, $C_2$ and $D$ are its arbitrary disjoint node sets. Then,*
*(i) $P(x_B|\sigma(\tilde{x}_{C_1}), \sigma(\tilde{x}_{C_2}), x_D) = P(x_B|\sigma(\tilde{x}_{C_2}), x_D)$ if $B \perp\!\!\!\perp_d C_1|D$ in $G^{\sigma(\tilde{x}_{C_2})}$;*
*(ii) $P(x_B|\sigma(\tilde{x}_{C_2}), x_D) = P(x_B|x_D)$ if there are no causal paths from $C_2$ to $B \cup D$ in $G$.*

**Proof of Lemma 14.** First, since $B \perp\!\!\!\perp_d C_1|D$ in $G^{\sigma(\tilde{x}_{C_2})}$, we know that $B \perp\!\!\!\perp_d C_1|D$ in $G^{\sigma(\tilde{x}_{C_1},\tilde{x}_{C_2})}$. Then, the result (i) follows by the fact that

$$P(x_B|\sigma(\tilde{x}_{C_1}), \sigma(\tilde{x}_{C_2}), x_D)$$
$$= P(x_B|\tilde{x}_{C_1}, \sigma(\tilde{x}_{C_2}), x_D) \quad \text{(by Rule 2 in Theorem 10)}$$
$$= P(x_B|\sigma(\tilde{x}_{C_2}), x_D) \quad \text{(by Rule 1 in Theorem 10)}.$$

Second, since there are no causal paths from $C_2$ to $B \cup D$, by Rule 3 in Theorem 11 we have

$$P(x_B, x_D|\sigma(\tilde{x}_{C_2})) = P(x_B, x_D) \text{ and } P(x_D|\sigma(\tilde{x}_{C_2})) = P(x_D),$$

which entail that the result (ii) holds. This completes all of the proofs.

**Proof of Theorem 10** (Rule 3). Let $C_1 = C \cap anc(D)$ and $C_2 = C/anc(D)$. It suffices to show

$$P(x_B|\sigma(\tilde{x}_{C_1}), \sigma(\tilde{x}_{C_2}), x_D) = P(x_B|w_D), \quad \text{if } B \perp\!\!\!\perp_d C|D \text{ in } G_{\overline{C_2}}. \tag{6}$$

First, we prove that if $B \perp\!\!\!\perp_d C|D$ in $G_{\overline{C_2}}$, then

$$B \perp\!\!\!\perp_d C_1|D \text{ in } G^{\sigma(\tilde{x}_{C_2})}, \tag{7}$$

and hence by Lemma 14(i) we have

$$P(x_B|\sigma(\tilde{x}_{C_1}), \sigma(\tilde{x}_{C_2}), x_D) = P(x_B|\sigma(\tilde{x}_{C_2}), x_D). \tag{8}$$

Suppose the result (7) does not hold. Then, there exists a $D$-connected path from $B$ to $C_1$ in $G^{\sigma(\tilde{x}_{C_2})}$. Note that this path can not contain any node in $C_2$. This is because if this path includes a node $c^* \in C_2$, then $c^* \notin anc(D)$ must be a collider, in view of the fact that the nodes $C_2$ in $G^{\sigma(\tilde{x}_{C_2})}$ have no output edges. It turns out that this path is blocked by $D$, leading to a contradiction. Therefore, since this $D$-connected path does not contain any node in $C_2$, it is also in $G_{\overline{C_2}}$, leading to a contradiction with the condition that $B \perp\!\!\!\perp_d C|D$ in $G_{\overline{C_2}}$

Second, we prove that if $B \perp\!\!\!\perp_d C|D$ in $G_{\overline{C_2}}$, then

$$\text{there are no causal paths from } C_2 \text{ to } B \text{ in } G, \tag{9}$$

and hence by Lemma 14(ii) and the fact that $C_2 \cap anc(D) = \emptyset$, we have

$$P(x_B|\sigma(\tilde{x}_{C_2}), x_D) = P(x_B|x_D). \tag{10}$$

Suppose the result (9) does not hold. Then, there exists a shortest causal path from $C_2$ to $B$ in $G$, and this shortest path contains only one node in $C_2$. Hence, this shortest path is also in $G_{\overline{C_2}}$. Since $B \perp\!\!\!\perp_d C|D$ in $G_{\overline{C_2}}$, it implies that $C_2 \cap anc(D) \neq \emptyset$, leading to a contradiction with the fact that $C_2 \cap anc(D) = \emptyset$.

Finally, the conclusion follows by (8) and (10).

**Proof of Theorem 12.** We first prove that if $B \perp\!\!\!\perp_d C|A, D$ in $G_{\overline{A}}$, then

$$B \perp\!\!\!\perp_d C|D \text{ in } G^{\sigma(\tilde{x}_A)}. \tag{11}$$

To prove (11), it suffices to show that any path $\ell$ from $B$ to $C$ in $G^{\sigma(\tilde{x}_A)}$ is blocked by $D$. We consider two different cases:

Case I: if the path $\ell$ contains a node $a^* \in A$, then $a^* \notin anc(D)$ must be a collider in $G^{\sigma(\tilde{x}_A)}$, since the nodes $A$ in $G^{\sigma(\tilde{x}_A)}$ have no output edges. Hence, the path $\ell$ is blocked by $D$ in Case I.

Case II: if the path $\ell$ contains no nodes in $A$, then $\ell$ is also a path in $G_{\overline{A}}$, and hence it is blocked by $A$ and $D$ in $G_{\overline{A}}$, due to the condition that $B \perp\!\!\!\perp_d C | A, D$ in $G_{\overline{A}}$. In other words, there exists a node $\kappa$, which blocks this path $\ell$ in $G_{\overline{A}}$. If $\kappa$ is a collider, then $\kappa \notin anc(A \cup D)$ in $G_{\overline{A}}$, indicating that there has no causal path from $\kappa$ to $D$ in $G_{\overline{A}}$. Then, it further implies that there has no causal path from $\kappa$ to $D$ in $G^{\sigma(\tilde{x}_A)}$, meaning that the path $\ell$ is blocked by $D$ in $G^{\sigma(\tilde{x}_A)}$.

If $\kappa$ is not a collider, then $\kappa \in A \cup D$ in $G_{\overline{A}}$. Since the path $\ell$ contains no nodes in $A$, it follows that $\kappa \in D$ in $G^{\sigma(\tilde{x}_A)}$, meaning that the path $\ell$ is blocked by $D$ in $G^{\sigma(\tilde{x}_A)}$.

Overall, we have shown that no matter whether $\kappa$ is a collider, the path $\ell$ is blocked by $D$ in Case II. Therefore, the result (11) holds. Similarly, we can show that if $B \perp\!\!\!\perp_d C | D$ in $G^{\sigma(\tilde{x}_A)}$, then $B \perp\!\!\!\perp_d C | A, D$ in $G_{\overline{A}}$. Hence, the result (i) holds.

Note that the nodes $C$ are chosen arbitrarily in the proof of (i). So, the results (ii)–(iii) follow by the same argument as for the result (i). This completes all of the proofs.

## Appendix B. Discussions on the Information Accounts of Causality

We might prefer not and think it's unnecessary to add more discussions about the philosophical accounts of causality in this paper. However, one of reviewers was expecting a description of the informational account of causality, and this reminds us that we might have the alternative option to present a short introduction and discussion about it.

The informational accounts of causality have been friendly introduced in detail by Chapter 13 of Illari and Russo (2014), and the core ideas are: 1) In causal inference, possible causes are ruled in and ruled out using what we know about possible causal links. But what is a causal link? 2) An informational account of causality holds that causal linking is informational linking. 3) Informational linking can be added to an account of linking by mechanisms, if mechanisms are seen as information channels. 4) A very broad notion of information can be used to capture all cases of causal linking, but further constraints could be added to the broad concept to characterize more narrow groups of cases of causal linking, such as in a particular scientific field. Several frequently asked questions are listed below:

Q1. John Collier was probably the first philosopher who tried to give an informational account of causality, what is the main idea?

A1. Briefly for collier, we can say causal connection is informational connection. Collier's idea is that the informational structure exists at every point in the process, and claimed that a major virtue of his theory is its generality.

Q2. Why information accounts for causal linking?

A2. Here we come to a place where philosophy explicitly borrows ideas from another field, in this case, from maths. Philosophical accounts want to use ideas from information theory because information theory is a very general language—and it is a language that was designed to allow us to characterize something like linking. So we shall make a quick detour to explain some key notions from information theory before returning to the philosophical theories.

Q3. How we understand mechanisms as causal links.

A3. Broadly, we find mechanisms that help us grasp causal linking in a coarse-grained way. Then we can think in terms of causal linking in a more fine-grained way by thinking informationally.

**Distinctions and warnings.** There are also some comparisons and warnings summarized in the book, including 1) If we are to understand reasoning about causal linking in diverse disciplines, we need a very generalized concept of a causal process or causal production. 2) Applicability: the major attraction of the informational account is its wide applicability. 3) Absences: an informational approach offers an entirely novel solution to the problem of absences, since it offers a genuinely revolutionary conception of linking. 4) Vacuity: an informational account may appear vacuous, but it is possible that the variety of informational concepts available will prove to be an advantage, rather than a problem.

Inspired by this accounts of causality, we introduced an intervention that intervening information transferred among variables, instead of intervening the variables themselves. Moreover, it is a more fine-grained way to think causality informationally, we might assume different causal graphs across individuals. If you want to learn more about philosophical accounts of causality, please read the book Illari and Russo (2014) and related materials.

