# OpenReview forum: "Info Intervention and its Causal Calculus"
_cclear.cc/CLeaR/2022/Conference — CLeaR 2022 Poster_

### Official Review · Reviewer_aWnE · 2021-11-21

**Confidence:** 2
**Overall Score:** 7

**Main Review:**

This paper proposes an info intervention framework based on structural causal model. Differing with the do-intervention, the info intervention has a different cutting strategy on DAG and passes the info of operated nodes to their descendants. The info intervention enjoys the same property of the do intervention, and the sigma operator brings convenience to annotate conditional probability of counterfactual variables. Overall, I think the paper is sound and vote for an acceptance.  Some typos: Fig references in Example 1 are missing.

**Summary:**

This paper proposes an info intervention framework based on structural causal model.

---

### Official Review · Reviewer_yRZ5 · 2021-11-23

**Confidence:** 3
**Overall Score:** 6

**Main Review:**

Originality: It is not entirely clear to me which contributions of the paper are novel. The authors note that graphically, their work is equivalent to SWIGs, and probabilistically, their work is equivalent to the po-calculus. Are the rules introduced in Section 4 novel, or do analogues already exist for the po-calculus? The info information comes with an operator, which is novel to this paper.

Significance: This paper will be interesting to anyone interested in the unification of ideas from the potential outcomes framework and the structural causal model framework, as well as anyone who is interested in framing causality in terms of transfer information.

Technical Quality: I have not read the proofs deeply, but the high-level ideas appear to be correct.

Clarity: There are several typos, grammatical errors, and broken links. There are several examples that are accompanied by figures, which are quite helpful.

For the same underlying SCM, are causal DAGs and info-causal dags identical? If so, I think it would be worth making this note.

I do not see where the graph subscript set overline terminology is defined (generally this notation is used to denote when arrows directed into the overline set have been removed from the graph).


**Summary:**

Summary: This paper proposes a concept the authors call the info intervention. The info intervention is motivated by the idea of transfer information and can be used as an alternative to the do intervention for reasoning about causal systems of variables. Additionally, they introduce the info-causal DAG which unifies ideas from the potential outcomes framework and the structural causal model framework. Ultimately, the authors derive a calculus for the info intervention which is similar to the do-calculus.

---

### Official Review · Reviewer_h6jK · 2021-11-27

**Confidence:** 3
**Overall Score:** 6

**Main Review:**

While I think the paper has some potentially interesting ideas, I also think it's not yet ready and honestly feels a bit rushed, since many parts are still unclear or missing.

For example, I was expecting a description of the informational account of causality from Collier (1999), how it differs from the other mainstream causality interpretations and how it relates to the new intervention, but this was never described. Moreover, it is just mentioned en passant that the info-causal DAGs are closely related to SWIGs and po-calculus, but it isn't really expanded upon.

# Details

I would write equation (2) in Definition differently to help see the difference with do-interventions, e.g. using the children of A

Markdown's latex mathmode is limited, but for example as a multiplication of:

$\Pi_{k \in ch(A)} P(x_k | x_{pa(k)\setminus A}, \tilde{x}_{pa(k) \cap A})$

$\Pi_{k \not \in ch(A)} P(x_k | x_{pa(k)})$

# Edit after rebuttal

While it seems the paper provides a theoretically sound approach, I think it might be a bit incremental/the novelty is not completely clear. This seems to be in line with what reviewer yRZ5 mentioned.

Despite the rebuttal claim that "assertions in the paper are quite formally clear and intuitive" it seems that at least reviewer yRZ5 and I had some questions about clarity and relation to previous work.

Since I still think it's interesting, I'm going to recalibrate my review to a weak accept (also in order to be more in line with the other reviewers), but I do encourage the authors to try to clarify better their work in the camera-ready, since it's also in their interest for the rest of the community to pick up on it.


**Summary:**

The paper presents a new type of intervention, the info intervention, which removes the outgoing edges of the intervention target. One of the motivations is about providing a calculus for an informational account of causality, with links to SWIGs and potential outcomes. The authors prove that with this new intervention they can derive a set of rules similar to the three do-calculus rules.

---

### Author Response · Authors · 2021-11-30
**Thanks for all the reviewers, we have a modified version pdf.**

Thanks for all the reviewers, we have a modified version pdf to address typos and other minor problems. Unfortunately, this is not ICLR which gives us a chance to upload a rebuttal version.

**For Reviewer h6jK:**  We think it's not necessary to add more discussions about the **philosophical view** of information accounts of causality, but still have added a introduction section  in **Appendix B**, and also the po-calculus is presented in the last paragraph of Section 4. We are actually a bit confused that **assertions in the paper are quite formally clear and intuitive**, but this reviewer still consider it as unclear or missing.  We still want to thank this reviewer for an alternative formation of equation (2), which though is not convenient for proofs in the Appendix A.

**For Reviewer yRZ5:** Thanks for you comments, all the minor problems you mentioned have been addressed in the updated pdf. The main concerns  Reviewer yRZ5 is "originality".  Our **novel framework** inspired by information accounts of causality,  owns a operator σ(·) for formalizing causal queries when compared with po-calculus with SWIGs, and it also owns the convenience of checking of independence between counterfactual variables when compared with do-intervention framework.

**Reviewer aWnE:** Thanks for you comments, typos have been resolved in our new version pdf.

---

### Decision · Program_Chairs · 2022-01-12

**Decision:**

Accept (Poster)

**Comment:**

This paper formulates the “info intervention” together with their calculus to formulate causal queries.

The paper is technically sound and interesting to unify ideas from the potential outcomes and the structural causal model frameworks. As such it will be of interest to the community.

Authors should improve the positioning of the work (for example in the concluding remarks), expanding the relations and differences with prior work and highlighting what is novel. This will be very useful for future work building on top of this new publication and was explicitly requested by reviewers as a condition to accept the paper.